# Non-Thermal Solar Wind Electron Velocity Distribution Function

**DOI:** 10.3390/e26040310

**Published:** 2024-03-30

**Authors:** Peter H. Yoon, Rodrigo A. López, Chadi S. Salem, John W. Bonnell, Sunjung Kim

**Affiliations:** 1Institute for Physical Science and Technology, University of Maryland, College Park, MD 20742, USA; 2Research Center in the Intersection of Plasma Physics, Matter, and Complexity (*P*^2^*mc*), Comisión Chilena de Energía Nuclear, Casilla 188-D, Santiago 7600713, Chile; rodrigo.lopez@cchen.cl; 3Space Sciences Laboratory, University of California, Berkeley, CA 94720, USA; salem@ssl.berkeley.edu (C.S.S.); jwbonnell@berkeley.edu (J.W.B.); 4Astronomy and Space Sciences, Kyung Hee University, Yongin 17104, Gyeonggi, Republic of Korea; sunjungkim1982@gmail.com

**Keywords:** kappa distribution, solar wind electrons, whistler-mode waves, turbulence, thermal fluctuations, electromagnetic, electrostatic, plasma, kinetic

## Abstract

The quiet-time solar wind electrons feature non-thermal characteristics when viewed from the perspective of their velocity distribution functions. They typically have an appearance of being composed of a denser thermal “core” population plus a tenuous energetic “halo” population. At first, such a feature was empirically fitted with the kappa velocity space distribution function, but ever since the ground-breaking work by Tsallis, the space physics community has embraced the potential implication of the kappa distribution as reflecting the non-extensive nature of the space plasma. From the viewpoint of microscopic plasma theory, the formation of the non-thermal electron velocity distribution function can be interpreted in terms of the plasma being in a state of turbulent quasi-equilibrium. Such a finding brings forth the possible existence of a profound inter-relationship between the non-extensive statistical state and the turbulent quasi-equilibrium state. The present paper further develops the idea of solar wind electrons being in the turbulent equilibrium, but, unlike the previous model, which involves the electrostatic turbulence near the plasma oscillation frequency (i.e., Langmuir turbulence), the present paper considers the impact of transverse electromagnetic turbulence, particularly, the turbulence in the whistler-mode frequency range. It is found that the coupling of spontaneously emitted thermal fluctuations and the background turbulence leads to the formation of a non-thermal electron velocity distribution function of the type observed in the solar wind during quiet times. This demonstrates that the whistler-range turbulence represents an alternative mechanism for producing the kappa-like non-thermal distribution, especially close to the Sun and in the near-Earth space environment.

## 1. Introduction

In situ measurements of charged particles in the near-Earth space environment by artificial satellite became possible during the decade of the 1960s. It was realized then that the velocity space distributions of charged particles that make up the space plasma deviate from the expected Maxwell–Boltzmann–Gauss statistics; instead, the observed distributions typically feature a suprathermal (or non-thermal) component with inverse power-law “tail” characteristics for the suprathermal velocity regime, $f\propto v^{-\gamma }$ for v≫α, where *v* represents the particle speed, *f* is the charged particle velocity distribution function, γ is the inverse power-law index, and α denotes the thermal speed [1,2,3]. Recent inner heliospheric missions, the Parker Solar Probe and Solar Orbiter, further confirm that such a non-thermal feature persists even for heliospheric environments much closer to the Sun [4,5,6]. The physical origin of such a feature was not understood then. Instead, Olbert and Vasyliunas [7,8,9,10] introduced an empirical model known as the kappa distribution,
(1)$$f_{\kappa }\left(v\right)\propto {\left(1+\frac{v^{2}}{\kappa \alpha^{2}}\right)}^{-(\kappa +1)},$$
to fit the observation. Here, $\alpha ={\left(2k_{B}T/m\right)}^{1/2}$ is the Maxwellian thermal speed, meaning that α is the thermal speed had f(v) been given by the Maxwell–Boltzmann distribution. $k_{B}=1.3806503\times 10^{-23}\;m^{2}\;\mathrm{kg}\;s^{-2}\;K^{-1}$ is the Boltzmann constant, which can replaced by unity if we adopt the unit of eV for thermal energy. That is, if the temperature *T* is expressed in eV instead of Kelvins (K), then we may take $k_{B}=1$. Hereafter, we shall adopt such a convention. The mass of the charged particles is denoted by *m*. The free parameter κ determines the degree to which the observed distribution deviates from the Maxwellian–Boltzmann (MB or thermal) distribution in that if κ→∞, then the model reduces to the thermal distribution, $f_{MB}\left(v\right)\propto \exp\left(-v^{2}/\alpha^{2}\right)$, while for v≫α, the kappa model depicts an inverse power-law velocity distribution, $f_{\kappa }\left(v\right)\propto v^{-2(\kappa +1)}$. It is to be noted that, regardless of the value of the κ index, the kappa distribution approximates the MB distribution for v≤α—to be more precise, for a low *v*, the kappa distribution approaches the MB distribution with a sightly lower thermal speed, by a factor of $\sqrt{\kappa /(\kappa +1)}$. That is, the kappa model naturally encompasses the quasi-Maxwellian feature in the “core” part of the velocity distribution characterized by v≤α and the inverse power-law tail portion of the distribution for the suprathermal regime, v≫α.

A sample non-thermal charged particle velocity distribution function in space is shown in Figure 1. Specifically, Figure 1 plots the typical electron velocity distribution function measured in the near-Earth space environment during quiet-time conditions. Figure 1 is a reproduction of Figure 4 of Ref. [11], and it shows two typical electron velocity distribution functions (eVDFs) in the solar wind at 1 au (astronomical unit) measured by the Wind/3DP electrostatic analyzers EESA-L and EESA-H. The left panels (a) and (c) show an eVDF in the slow solar wind (at 1995-06-19/00:06:38), and the right panels (b) and (d) show an eVDF in the fast solar wind (at 1995-06-19/23:13:59). The top panels (a) and (b) show cuts through the eVDF in one of the two directions perpendicular to the local magnetic field B: the diamonds are data points from EESA-L and the asterisks are data points from EESA-H. The dotted lines represent the one-count level for EESA-L and EESA-H. The blue dashed line in Figure 1a,b represents the sum of Maxwellian and kappa distributions (indicated in blue). The red line represents the fit to the measured perpendicular eVDF cut; the resulting fit parameters are indicated in red. The bottom panels (c) and (d) show cuts through the eVDF in the direction parallel to B. The perpendicular fit is shown in red, and the perpendicular fit parameters are used to initialize the parallel eVDF fit. The blue dashed line in Figure 1a,b represents the sum of Maxwellian and kappa distributions calculated using independent measurements of the core and halo densities and temperatures obtained from the fit of the spectrum of quasi-thermal fluctuations around the electron plasma frequency measured by the Wind/Waves electric field antennas. This “quasi-thermal noise” (QTN) technique is immune to spacecraft potential and therefore offers an independent and highly accurate measure of the core electron density and temperature, which are used as a reference to initiate the nonlinear least squares fitting of the measured VDF, resulting in the red curve fit, whose fit parameters are indicated in red as well. For more details, see Ref. [11] and Figure 4 therewith, including the accompanying description.

It is well known that the MB distribution corresponds to the maximum entropic (or the most probable) state as defined through the textbook Boltzmann–Gibbs (BG) definition for the entropy [12,13,14], namely,
$$S_{BG}=-k_{B}\int dx\int dvf\left(v\right)\ln f\left(v\right),$$
where ∫dx is the spatial integration normalized to the total volume, $\int dx\to V^{-1}\int dx$, and f(v) is the velocity distribution function. The Boltzmann–Gibbs (BG) entropy, which is additive and extensive, applies to an ideal gas or systems dictated by short-range interactions. The suitability of BG entropy for systems interacting through long-range forces, such as the plasma or gravitational systems, has been questioned since the inception of the BG entropy in the first place [15,16,17]. The additive property relates to the BG entropy of a total system being equal to the entropies of subsystems. The extensivity means that the entropy is proportional to the total number of particles. The non-additive/non-extensive entropy, which presumably may be applicable to systems governed by long-range forces, violates these properties [18]. The mathematical form of non-extensive entropy, which became well-known thanks to the work by Tsallis [19], was apparently independently discovered several times over, as entry 107 in Ref. [18], p. 347, describes. Specifically, it is mentioned there that several authors have independently rediscovered the form of entropy
$$S_{q}=k\;\frac{1-\sum_{i=1}^{W}p_{i}^{q}}{q-1}.$$
The list includes J. Havrda and F. Charvat, Kybernetika **3**, 30 (1967); I. Vajda, Kybernetika **4**, 105 (1968); Z. Daroczy, Inf. Control **16**, 36 (1970); J. Lindhard and V. Nielsen, *Studies in statistical mechanics*, Det Kongelige Danske Videnskabernes Selskab Matematisk-fysiske Meddelelser (Denmark) **38** (9), 1 (1971); B. D. Sharma and D. P. Mittal, J. Math. Sci. **10**, 28 (1975); J. Aczel and Z. Daroczy, *On Measures of Information and Their Characterization*, in *Mathematics in Science and Engineering,* ed. R. Bellman (Academic Press, New York, 1975); A. Wehrl, Rev. Mod. Phys. **50**, 221 (1978); and G. P. Patil and C. Taillie, An overview of diversity, in *Ecological Diversity in Theory and Practice*, eds. J. F. Grassle, G. P. Patil, W. Smith, and C. Taillie (Int. Cooperat. Publ. House, Maryland, 1979), pp. 3–27.

These earlier works notwithstanding, it is Tsallis’s model [19] that is most well known, and it has triggered an explosive growth of recent interest in the topic of non-extensive thermostatics, in the space plasma context as well as in other applications [10,20,21]. The celebrated Tsallis entropy in continuum form is defined by
$$S_{q}=-\frac{k_{B}}{1-q}\int dx\int dv\left\{f\left(v\right)-{\left[f\left(v\right)\right]}^{q}\right\},$$
and the velocity distribution that corresponds to the maximum entropic (or the most probable) state is given by
(2)$$f_{q}\left(v\right)\propto {\left(1+\frac{\left(1-q\right)\;v^{2}}{\alpha^{2}}\right)}^{-1/(1-q)}.$$
Upon identifying κ=1/(1−q) or alternatively κ=q/(1−q), one finds that the solution reduces to either $f∼{\left[1+v^{2}/\left(\kappa \alpha^{2}\right)\right]}^{-\kappa }$ or $f∼{\left\{1+v^{2}/\left[\left(\kappa +1\right)\alpha^{2}\right]\right\}}^{-\kappa -1}$, respectively. Strictly speaking, neither is exactly identical to the kappa distribution since $f_{\kappa }$ is defined with a mixed κ and κ+1—see Equation (Equation 1). Nonetheless, this convergence of Tsallis’s non-extensive entropic principle and the kappa model has led to the space physics community embracing the notion that the space plasma may be in a state of non-extensive statistical quasi-equilibrium [10,20,22,23,24].

From the microscopic plasma physics, it is known that the electron kappa distribution can be regarded as an end product of the weak electrostatic Langmuir turbulence [25,26]. The initial findings involved a numerical study of a gentle weak electron beam–plasma (or bump-on-tail) instability and subsequent saturation of the Langmuir turbulence. It was found that the quasi-steady state of the Langmuir turbulence is characterized by the formation of a non-thermal, kappa-like velocity distribution function. Subsequently, more rigorous theoretical analysis revealed that the kappa distribution belongs to a family of unique solutions that characterize a steady-state electrostatic plasma turbulence [27,28]. This finding implies that a profound inter-relationship may exist between the non-extensive statistical state and the turbulent quasi-equilibrium, but the precise mathematical formulation to establish such a connection does not yet exist at present.

The findings in Refs. [27,28] directly relate to the solar wind electrons [11,29], which can be interpreted as velocity distribution functions made of multiple subcomponents. The primary component is the quasi-Maxwellian core population (∼90–95% of the density, with ∼10 eV). The hotter and tenuous halo electron population can be distinguished from the core population by its distinct velocity profile, which can be modeled by an inverse power law. Other distinct populations can also be identified. For high-speed solar wind streams, a highly field-aligned strahl component can be separately classified from the halo electrons by their narrow pitch-angle distribution. The halo/strahl density is about ∼5–10% of the total density with a ∼50 eV energy range. Also, the highly energetic superhalo electrons (with a typical energy in the ∼2 keV range but extending up to 100 keV), which are observed in nearly all solar wind conditions, including the inner heliosphere [4,5] with a nearly invariant velocity power-law index, are a distinct component [29,30,31]. The core, halo/strahl, and superhalo electron populations are sometimes associated with their respective slight temperature anisotropy, although the superhalo is almost completely isotropic, and relative drifts between the core and halo can also be detected. In the present discussion, however, we idealize the situation by considering that the velocity distribution is isotropic and without any net drifts.

In the present paper, we will first briefly overview the previous weak turbulence theory of electron kappa distribution [27], but, thereafter, we will discuss a new development, which involves the whistler-mode fluctuations and turbulence. For the near-Earth space plasma environment as well as for the inner heliosphere close to the Sun, the effects of wave–particle resonant interaction that involves the whistler-mode waves, instability, and fluctuations on the electrons are important [6,32,33,34,35]. As such, we consider the consequence of the electrons undergoing wave–particle resonant interactions with the background turbulence in the whistler-mode frequency range in the present paper. As will be shown, the impact of such interactions is none other than the formation of a non-thermal velocity distribution function for the electrons, which is not necessarily the kappa distribution but rather a more general one that must be generated by a numerical indefinite velocity integration. However, in the theoretical formalism of the present paper, it turns out that thermal fluctuations play an important role. A finite-temperature plasma constantly spontaneously emits and reabsorbs electromagnetic fluctuations—the fluctuation–dissipation theorem. A correct self-consistent theory of steady-state plasma particle velocity distribution based upon the steady-state Fokker–Planck particle kinetic equation thus requires the computation of thermal fluctuations. We thus begin the discourse by considering the thermal fluctuations emitted by the core electrons and the modification of the fluctuation spectrum by the presence of background turbulence.

## 2. Thermal Fluctuations Emitted by Maxwellian Core Electrons in the Background of Solar Wind Turbulence

In this section, we discuss the quasi-steady-state spectrum of the electrostatic and electromagnetic fluctuations in the background of solar wind turbulence. We assume that the thermal fluctuations are spontaneously emitted and reabsorbed predominantly by the Maxwellian core electrons. The background large-amplitude turbulence is assumed to be of the transverse electromagnetic type, with its characteristic frequency that encompasses the whistler-mode frequency range. The combined fluctuations and turbulence spectra determine the quasi-steady-state velocity distribution function for the solar wind halo electrons. As already discussed, the solar wind electrons are observed to be made of several distinct components, but the simplest description pertains to the two-component model, in which these electrons comprise dense Maxwellian core electrons and a tenuous but energetic halo electron population. It turns out that the halo electrons immersed in the field of thermal fluctuations alone will be organized in velocity space into a Maxwellian distribution. Thus, in this case, there will be no distinction between the core and halo so that both species will form one continuous thermal population. However, if there exist turbulent wave spectra for the whistler mode, then as the electrons interact with these combined spontaneously generated fluctuations and turbulence, they will organize into a non-thermal velocity distribution function, which manifests a clear demarcation between the core population and a tail component. The spontaneous emission is important because these background fluctuations provide the basis upon which non-thermal distribution can be built.

The electrostatic component of the spontaneous emission [36,37] is the well-known quasi-thermal noise [38], but the solar wind core electrons should also emit electromagnetic emissions as well, although a clear identification of such a transverse quasi-thermal noise is difficult because it will be partially occulted by the background turbulence. However, with improved future detection techniques, identifying the transverse quasi-thermal noise may become possible. Although we expect the electric and magnetic fields associated with the whistler-mode fluctuations to partially overlap with the frequency range of the background solar wind turbulence, the spectrum should extend to slightly higher frequencies so that with sufficiently sensitive instruments, the identification could be possible. Even with today’s technology, if one analyzes the data with sufficient accuracy for the high-frequency end of the spectrum, one should be able to discern the characteristic signature associated with the whistler-mode thermal spectrum. Regardless, from a theoretical perspective, consideration of the emission of electromagnetic fluctuations in the whistler mode is important. In the presence of the combined background spectrum of Langmuir and whistler-mode fluctuations as well as the whistler wave turbulence, it will be shown that the solar wind electrons naturally form a non-thermal velocity distribution function of the type observed in space, but it is not necessarily the kappa model in the analytic sense. Rather, the model distribution will be obtained by a numerical indefinite velocity integration.

The first step in the present discussion is to consider the spectrum of electrostatic and electromagnetic fluctuations emitted by the thermal core electrons. In Ref. [39], the formulae for these fluctuation spectra are derived. For electromagnetic fluctuations propagating in a parallel direction with respect to the ambient magnetic field vector, the transverse electric and magnetic field spectra are designated as ${\langle \delta E_{⊥}^{2}\rangle }_{k,\omega }$ and ${\langle \delta B_{⊥}^{2}\rangle }_{k,\omega }$, while for electrostatic fluctuations characterized by propagation parallel to the ambient magnetic field, the electric field spectrum is denoted by ${\langle \delta E_{\|}^{2}\rangle }_{k,\omega }$. These are given by [39,40,41].
(3)$$\begin{array}{ccc} {\langle \delta E_{⊥}^{2}\rangle }_{k,\omega } & = & \frac{2k_{m}^{2}e^{2}}{\omega^{2}{\left|\epsilon_{⊥}\left(k,\omega \right)-c^{2}k^{2}/\omega^{2}\right|}^{2}}\int dv\;v_{⊥}^{2}\;\delta \left(\omega -kv_{\|}-\Omega_{e}\right)f,\\ {\langle \delta B_{⊥}^{2}\rangle }_{k,\omega } & = & \frac{c^{2}k^{2}}{\omega^{2}}\;{\langle \delta E_{⊥}^{2}\rangle }_{k,\omega },\\ {\langle \delta E_{\|}^{2}\rangle }_{k,\omega } & = & \frac{2k_{m}^{2}e^{2}}{k^{2}{\left|\epsilon_{\|}\left(k,\omega \right)\right|}^{2}}\int dv\;\delta \left(\omega -kv_{\|}\right)f,\\ \epsilon_{⊥}\left(k,\omega \right) & = & 1+\frac{\omega_{pe}^{2}}{\omega^{2}}\int dv\;\frac{v_{⊥}/2}{\omega -kv_{\|}-\Omega_{e}}\left(\left(\omega -kv_{\|}\right)\;\frac{\partial f}{\partial v_{⊥}}+kv_{⊥}\frac{\partial f}{\partial v_{\|}}\right),\\ \epsilon_{\|}\left(k,\omega \right) & = & 1+\frac{\omega_{pe}^{2}}{k}\int dv\;\frac{\partial f/\partial v_{\|}}{\omega -kv_{\|}}, \end{array}$$
where *e* is the unit electric charge; $\omega_{pe}={\left(4\pi n/m_{e}\right)}^{1/2}e$ is the plasma frequency, with *n* and $m_{e}$ being the ambient density and electron mass, respectively; $\Omega_{e}=eB/\left(m_{e}c\right)$ is the electron cyclotron frequency, with *B* and *c* being the ambient magnetic field intensity and the speed of light, respectively; and $k_{m}^{2}=\Omega_{e}^{2}/\alpha_{e}^{2}$ is the maximum perpendicular wave length, which results from the integration over the perpendicular wave number, with $\alpha_{e}={\left(2T/m_{e}\right)}^{1/2}$ being the electron thermal speed [40]. Here, $f=f(v_{⊥},v_{\|})$ represents the electron velocity distribution function (normalized to unity, ∫dvf=1), with $v_{⊥}$ and $v_{\|}$ denoting the velocity component perpendicular and parallel to the ambient magnetic field. The angular frequency and the parallel wave number are defined by ω and *k*, respectively.

For the Maxwellian thermal velocity distribution function, these are given as shown below:(4)$$\begin{array}{ccc} {\langle \delta E_{⊥}^{2}\rangle }_{k,\omega } & = & \frac{\omega_{pe}^{2}k_{m}^{2}T_{e}e^{-\zeta^{2}}}{4\pi^{3/2}\omega^{2}k\alpha_{e}{\left|\epsilon_{⊥}\left(k,\omega \right)-c^{2}k^{2}/\omega^{2}\right|}^{2}},\;{\langle \delta B_{⊥}^{2}\rangle }_{k,\omega }=\frac{c^{2}k^{2}}{\omega^{2}}\;{\langle \delta E_{⊥}^{2}\rangle }_{k,\omega },\\ {\langle \delta E_{\|}^{2}\rangle }_{k,\omega } & = & \frac{\omega_{pe}^{2}k_{m}^{2}T_{e}e^{-\xi^{2}}}{\pi^{3/2}k^{3}\alpha_{e}^{3}{\left|\epsilon_{\|}\left(k,\omega \right)\right|}^{2}},\\ \epsilon_{⊥}\left(k,\omega \right) & = & 1-\frac{c^{2}k^{2}}{\omega^{2}}+\frac{\omega_{pe}^{2}}{\omega^{2}}\;\xi Z\left(\zeta \right),\;\zeta =\frac{\omega -\Omega_{e}}{k\alpha_{e}},\\ \epsilon_{\|}\left(k,\omega \right) & = & 1-\frac{\omega_{pe}^{2}}{k^{2}\alpha_{e}^{2}}\;Z^{\prime }\left(\xi \right),\;\xi =\frac{\omega }{k\alpha_{e}}, \end{array}$$
where $Z\left(\zeta \right)=\pi^{-1/2}\int_{-\infty }^{\infty }dxe^{-x^{2}}{\left(x-\zeta \right)}^{-1}$, Im(ζ)>0, is the plasma dispersion function with the prime indicating the derivative with respect to the argument.

Figure 2 plots the electrostatic and electromagnetic spectra, ${\langle \delta E_{\|}^{2}\rangle }_{k,\omega }$ and ${\langle \delta E_{⊥}^{2}\rangle }_{k,\omega }$, respectively, computed from the theoretical formulae (Equation 4), versus $ck/\omega_{pe}$ (horizontal axis) and $\omega /\Omega_{e}$ (vertical axis). The color scale is relative in that the maximum value for each panel is represented by red and the minimum intensity is plotted as a blue backdrop. The left-hand top and bottom panels correspond to the electrostatic and electromagnetic fluctuation spectra, respectively. The input parameters are $\omega_{pe}/\Omega_{e}=5$ and $\beta_{e}=1$, where $\beta_{e}=8\pi nT_{e}/B^{2}$ is the electron beta (ratio of electron thermal energy to the magnetic field energy). In order to verify that the theoretical formalism (Equation 4) is indeed reliable, we have also carried out a one-dimensional particle-in-cell (PIC) simulation. We have used a simulation box of $L_{x}=512\;c/\Omega_{pe}$ and $n_{x}=4096$ grid points, with 2000 particles per grid per species. The time step used was $\Delta t=0.01\;/\omega_{pe}$, and the simulation ran until $t=2621.44\;/\omega_{pe}$. The ratio of plasma frequency to electron gyro frequency was $\omega_{pe}/\Omega_{e}=5$. This ratio is somewhat lower than the actual value typical of the solar wind at 1 au, which is close to $\omega_{pe}/\Omega_{e}∼O\left(10\right)$–$O(10^{2})$, but for the sake of illustration we have chosen a relatively low value of $\omega_{pe}/\Omega_{e}$. Otherwise, the spectral peak at $\omega ∼\omega_{pe}$ associated with the electrostatic thermal fluctuations (upper panels) and the spectral characteristics associated with the transverse-mode fluctuations around the electron cyclotron frequency and below would have been separated by a wide gap, which would have made visual inspection quite challenging. Also, if the separation between the two frequencies is too high, it becomes very challenging for the simulations, too, because we need to resolve both time scales. Other parameters were electron and proton betas, which were taken to be $\beta_{e}=\beta_{p}=0.1$. These choices are not atypical of the solar wind conditions at 1 au. The simulated electrostatic and electromagnetic fluctuation spectra are plotted in the top and bottom right-hand panels, respectively. As the readers may appreciate, the theoretical plots compare very well with the simulated spectra, which indicates that the theoretical method is a reliable tool for describing the spontaneously emitted thermal spectra in magnetized plasmas accurately.

The electrostatic fluctuation spectrum is enhanced along the Langmuir wave dispersion curve but broadens in frequency somewhat for shorter wavelengths. In the simulated spectrum, the enhanced fluctuation along the Langmuir wave dispersion curve is broader than that of the theoretical spectrum, but, otherwise, the overall agreement is excellent. For the electromagnetic spectrum, it is seen that the fluctuation spectrum is enhanced along the whistler-mode dispersion curve, but the triangular (or conical) emission pattern that converges to the electron cyclotron frequency, $\omega =\Omega_{e}$ at the k→0 limit, is also prominent in both the theoretical emission spectrum and the simulated spectrum. Such a feature is associated with the virtual (or higher-order) modes, that is, heavily damped solutions of the linear dispersion relation [42,43,44]. Both the theoretical and simulated spectra accurately reproduce the emission characteristics associated with such modes. Note, however, that the simulation does not completely demonstrate the intensification of the higher-order mode as *k* approaches a zero value. This is owing to the limited resolution in the simulation spectrum. As will be discussed, this limitation further affects the *k*-integrated wave spectra for the electric and magnetic fields.

Shown in Figure 3 are wave number-integrated (∫dk⋯) spectra. The left-hand panel shows the *k*-integrated magnetic and transverse electric field fluctuation spectra $\int dk{\langle \delta B^{2}\rangle }_{k,\omega }$ (red) and $\int dk{\langle \delta E_{⊥}^{2}\rangle }_{k,\omega }$ (blue) that were computed from theory, plotted against $\omega /\Omega_{e}$. The right-hand panel displays the same spectra constructed from the PIC simulation result and integrated over the wave numbers. Both the theoretical and simulated spectra exhibit the behavior of increasing intensities, for both magnetic and electric spectra, over an increasing frequency, up to $\omega ∼0.5\Omega_{e}$ or so. However, some differences in the behavior are also evident. For instance, in the theoretical integrated spectra, both the electric and magnetic field intensify around the electron cyclotron frequency, $\omega ∼\Omega_{e}$, but the simulated spectra do not exhibit such a behavior. Clearly, the peak at $\omega /\Omega_{e}=1$ in the theoretical spectrum is associated with the contribution from the higher-order mode. In the simulated spectrum, the higher-order mode for the low *k* regime is not as clearly enhanced, which explains the absence of such a peak. This is due to the limited resolution in the simulation. Such an increasing behavior as a function of frequency for the fluctuation spectra in the low-frequency regime is a characteristic of the plasma, and it is the baseline spectral behavior associated with the thermal motion of plasma particles. It is interesting to note that in many PIC simulations of low-frequency turbulence, such an increasing intensity can be seen at the high end of the simulation spectrum. In a typical kinetic simulation of the low-frequency turbulence, the MHD-like regime corresponding to $\omega^{2}\ll \Omega_{p}^{2}\ll \Omega_{e}^{2}$, where $\Omega_{i}=eB/\left(m_{p}c\right)$ is the proton cyclotron frequency, is characterized by a Kolmogorov type of inverse power-law spectrum, $k^{-5/3}$ [45,46,47], but as *k* increases, in some cases, the intensity actually rises again [47,48]. In the literature, such a behavior is not clearly explained nor understood. However, it is entirely possible that the simulation system is automatically generating the background thermal spectrum.

As confirmed by Figure 2 and Figure 3, the theoretical description of thermal fluctuations is consistent with the simulation result. Thus, we now focus on the analytical approach. Furthermore, henceforth, we are interested in the fluctuations associated with the eigen modes. For the electrostatic fluctuation, we are concerned with the spectral wave intensity along the Langmuir mode dispersion relation, $\omega =\omega_{L}\left(k\right)$, where $\omega_{L}=\omega_{pe}\left[1+3k^{2}\alpha_{e}^{2}/\left(4\omega_{pe}^{2}\right)\right]$. Likewise, for the electromagnetic fluctuations, we pay attention to the whistler-mode dispersion relation, $\omega =\omega_{W}\left(k\right)$, where $\omega_{W}=\Omega_{e}c^{2}k^{2}/\left(\omega_{pe}^{2}+c^{2}k^{2}\right)$. Then, by expanding the denominators by ϵ‖≈(ω−ωL+i0)(∂Reϵ‖/∂ωL)+iImϵ‖ and Λ+≈(ω−ωW+i0)(∂ReΛ+/∂ωW)+iImΛ+ while ignoring the contribution from the term associated with $\Lambda_{-}$, it is possible to obtain
(5)$$\begin{array}{ccc} {\langle \delta E_{\|}^{2}\rangle }_{k,\omega } & = & I_{L}\left(k\right)\;\delta \left(\omega -\omega_{L}\right),\\ {\langle \delta E_{⊥}^{2}\rangle }_{k,\omega } & = & I_{W}\left(k\right)\;\delta \left(\omega -\omega_{W}\right),\;{\langle \delta B_{⊥}^{2}\rangle }_{k,\omega }=M_{W}\left(k\right)\;\delta \left(\omega -\omega_{W}\right), \end{array}$$
where
(6)$$\begin{array}{ccc} I_{L}\left(k\right) & = & \frac{k_{m}^{2}T_{e}}{4\pi },\\ I_{W}\left(k\right) & = & \frac{k_{m}^{2}T_{e}}{4\pi }\frac{\omega_{pe}^{2}\omega_{W}^{3}}{c^{4}k^{4}}=\frac{k_{m}^{2}T_{e}}{4\pi }\frac{\Omega_{e}^{2}\omega_{pe}^{2}c^{2}k^{2}}{{\left(\omega_{pe}^{2}+c^{2}k^{2}\right)}^{3}},\\ M_{W}\left(k\right) & = & \frac{k_{m}^{2}T_{e}}{4\pi }\frac{\omega_{pe}^{2}\omega_{W}}{c^{2}k^{2}}=\frac{k_{m}^{2}T_{e}}{4\pi }\frac{\omega_{pe}^{2}}{\omega_{pe}^{2}+c^{2}k^{2}}. \end{array}$$
For more details regarding the derivation of this result, see [39,40,41].

In the solar wind, there exists a permanent low-frequency turbulence of a solar origin. Such turbulence is commonly believed to be generated on the surface of the Sun through various mechanisms, including the solar surface convection and small reconnection near the lower corona, and convected to outer space [49]. The solar wind turbulence for a low-MHD frequency regime is hydromagnetic in nature and is characterized by a Kolmogorov-like inertial range spectrum but with a spectral break in the kinetic regime. That is, for the frequency range above the nominal proton cyclotron frequency and below the electron cyclotron frequency, $\Omega_{p}<\omega <\left|\Omega_{e}\right|$, the turbulence exhibits a spectral break. Such a frequency range can be characterized as the whistler turbulence range. For an even higher frequency $\omega >|\Omega_{e}|$, another spectral break is present. We may model such a multi-scale spectral behavior by adopting an analytical model first suggested by von Kármán [50] and generalizing to reflect the multiple spectral breaks,
(7)$$I_{\mathrm{turb}}\left(k\right)=\frac{k_{m}^{2}I_{0}}{{\left(1+k^{2}l^{2}\right)}^{\alpha }{\left(1+c^{2}k^{2}/\omega_{pi}^{2}\right)}^{\beta -\alpha }{\left(1+c^{2}k^{2}/\omega_{pe}^{2}\right)}^{\gamma -\beta }},$$
where $l\gg c/\omega_{pi}$. Here, we explicitly extracted out the factor $k_{m}^{2}$ since this is related to the integration over $k_{⊥}$ [40]. The solar wind turbulence spectrum appears to behave as $\omega^{-5/3}$ in the frequency range corresponding to the MHD regime. If we make use of the Taylor hypothesis [51], then ω can be trivially replaced by *k*, but in the kinetic regime, beyond the ion skin depth, $c/\omega_{pi}$ or shorter, and much more so for the electron skin depth, $c/\omega_{pe}$ or shorter, the Taylor hypothesis may not be valid. Moreover, since we are interested in the parallel wave vector and the turbulence intensity integrated over $k_{⊥}$, the inverse power-law index α may not be the same as that of the Kolmogorov value, namely α=5/6. Nevertheless, we may model the MHD regime by the Kolmogorov type of spectrum. In any event, the model spectrum (Equation 8) describes a finite and maximum turbulence level at k=0, and for $0<k<\omega_{pi}/c$ it describes the $k^{-2\alpha }$ behavior. For the wave number regime corresponding to $\omega_{pi}/c<k<\omega_{pe}/c$, the model depicts a $k^{-2\beta }$ behavior. For $k>\omega_{pe}/c$, the spectrum behaves as $k^{-2\gamma }$. We illustrate this by choosing α=5/6, β=1.2, and γ=2, which are admittedly arbitrary. We also choose the MHD scale factor $l=10^{2}\left(m_{p}/m_{e}\right)$, which is again arbitrary.

In Figure 4, we demonstrate the influence of the whistler-mode fluctuation spectrum on the background turbulence spectrum by considering the superposition of the model turbulence spectrum and the whistler-mode fluctuation spectrum, $I_{\mathrm{turb}}\left(k\right)+I_{W}\left(k\right)$, where the whistler-mode fluctuation spectrum $I_{W}\left(k\right)$ is defined in Equation (Equation 6) and the model turbulence spectrum $I_{\mathrm{turb}}\left(k\right)$ is given by Equation (Equation 7). Figure 4 plots the spectral factor that defines the combined spectrum, namely,
(8)$$\begin{array}{ccc} S(q) & = & \frac{q^{2}}{{\left(1+q^{2}\right)}^{3}}+\frac{R}{{\left(1+Mlq^{2}\right)}^{\alpha }{\left(1+Mq^{2}\right)}^{\beta -\alpha }{\left(1+q^{2}\right)}^{\gamma -\beta }},\\ q & = & \frac{ck}{\omega_{pe}},\;M=\frac{m_{p}}{m_{e}}=1836,\;R=\frac{4\pi I_{0}}{T_{e}}\frac{\omega_{pe}^{2}}{\Omega_{e}^{2}}, \end{array}$$
where α=5/6, β=1.2, γ=2, and $l=10^{2}\left(m_{p}/m_{e}\right)$, as already noted above. The first term on the right-hand, $q^{2}/{\left(1+q^{2}\right)}^{3}$, denotes the spontaneously emitted whistler-mode thermal fluctuation spectrum. The second term on the right-hand side is the model spectrum with multiple spectral breaks. In Figure 4, the dashed magenta-colored curve represents the spontaneously emitted whistler-mode fluctuation spectrum, $q^{2}/{\left(1+q^{2}\right)}^{3}$. The dashed black curves represent the background turbulence spectrum without the influence of the fluctuation, for two cases of $R=10^{6}$ and $R=10^{7}$. The total spectral factor for the two cases is plotted with thick blue ($R=10^{6}$) and red ($R=10^{7}$) curves. It is evident that the model turbulence spectrum (Equation 8) depicts a flat spectrum for a q→0 regime; a Kolmogorov-type of spectrum in the “MHD” regime, ${\left(Ml\right)}^{-1/2}<q<M^{-1/2}$; a slightly steeper spectrum of $k^{-2.4}$ in the “kinetic proton” regime, $M^{-1/2}<q<1$; and a yet steeper spectrum, $k^{-4}$, in the whistler turbulence regime, q>1. It is in this wave number regime where the presence of the thermal fluctuation spectrum should be discernible. Specifically, in the case of a relatively low turbulence level, as indicated by $R=10^{6}$, we expect that the actual solar wind turbulence should reveal the presence of the fluctuation. However, for higher turbulence levels (as denoted by $R=10^{7}$), the intensity of fluctuation will be partially hidden so that a clear identification might not be so straightforward.

As an example of actual solar wind turbulence spectra measured in the near-Earth environment, we reproduce a figure taken from Ref. [52]. The result is Figure 5, which is constructed from the measurements made by *Cluster* spacecraft. The location of the spacecraft is at 1 au during a quiet-time condition on 30 January 2003. The detailed discussion of the instrumentation and data analysis method can be found in Ref. [52], but the main focus of the present paper is bring the readers’ attention to the spectral flattening behavior for the high-frequency end of the turbulence spectra, especially for the electric field. According to the theory—see Equation (Equation 6)—and the model spectrum shown in Figure 4, the spontaneously emitted thermal fluctuations should affect the high end of the solar wind turbulence spectra, especially if the turbulence level is sufficiently low. Admittedly, just what exactly it means by “sufficiently low” is not entirely clear, and further study is called for. Nevertheless, the identification of the spontaneous quasi-thermal whistler-mode fluctuations based on observation could be an intriguing and innovative research topic. In any case, Figure 5 displays the Kolmogorov-like spectrum in the low-frequency band while also showing a spectral break at frequency $f_{b}$, which represents the “break” frequency for the transition of one spectral slope to another. This frequency could be associated with the kinetic proton effects. The whistler-mode thermal noise, however, is supposed to be associated with the electron kinetic effects, which are believed to be related to a much higher frequency. However, before one could reach such a frequency, the instrument noise floor would contaminate the data, so it is very challenging to delineate the noise effects versus the baseline thermal noise.

Specifically, a key element that should be accounted for before one can definitely extract the theoretical signature, i.e., the whistler-mode thermal fluctuation, from the data is for the model turbulence spectrum of the type shown in Figure 4 to be translated into the spacecraft frame frequency using the appropriate solar wind speed and electron inertial length, as well as to properly scale the normalized amplitude S(q) to physical units. This includes translating the “*R*” parameter into actual units. The flattening of the *E*-field spectrum shown in Figure 5 could be entirely due to the instrumental artifact. In spite of this, however, the thermal noise associated with the whistler-mode fluctuations could partly contribute to the observed flattening of the spectrum, if not for this particular event, then at least for some other events. As will be shown in the next section, the combination of the quasi-thermal whistler noise and the background turbulence can account for the observed non-thermal electron velocity distribution function. We thus proceed with the discussion of the theory for the formation of the electron velocity distribution function under the influence of background whistler-mode turbulence and the quasi-thermal noise spectrum, which contains both electrostatic Langmuir-type and whistler-mode-type electromagnetic fluctuations.

## 3. Formation of Kappa Electron Distribution by Langmuir Turbulence

In this section, we briefly overview the previous theory of kappa electron distribution by Langmuir turbulence advanced by Yoon [27]. The full discourse of this theory is quite complex and requires a detailed exposition of kinetic weak plasma turbulence theory [37,53,54,55,56,57,58,59,60,61,62,63,64,65], but, in its essence, it boils down to the modification of the spontaneously emitted Langmuir fluctuations to reflect the influence of the steady-state weak Langmuir turbulence spectrum. It was shown by considering the balance of the nonlinear wave kinetic equation for Langmuir turbulence that, in the steady state, the electrostatic fluctuation spectrum should be modified to include the effects of turbulence in the following form:(9)$$I_{L}\left(k\right)=\frac{k_{m}^{2}T_{e}}{4\pi }\left(1+\frac{k_{L}^{2}}{k^{2}}\right).$$
The modification factor $k_{L}^{2}/k^{2}$ leads to the kappa electron distribution function when this spectrum is inserted into the diffusion coefficient of the steady-state electron distribution function computed from the kinetic theory.

Reference [40] derives the Fokker–Planck electron kinetic equation with waves and fluctuations that have a wave vector lying in the parallel direction defined with the ambient magnetic field vector. We summarize the equation for the electron velocity distribution function *f*,
(10)$$\begin{array}{ccc} \frac{\partial f}{\partial t} & = & \frac{1}{v^{2}}\frac{\partial }{\partial v}\;\left(v^{2}A_{v}f\right)+\frac{1}{v}\frac{\partial }{\partial \mu }\;\left(A_{\mu }f\right)\\ & & +\frac{1}{v^{2}}\frac{\partial }{\partial v}\left(v^{2}D_{vv}\frac{\partial f}{\partial v}\right)+\frac{1}{v^{2}}\frac{\partial }{\partial v}\left(vD_{v\mu }\frac{\partial f}{\partial \mu }\right)\\ & & +\frac{1}{v}\frac{\partial }{\partial \mu }\left(D_{v\mu }\frac{\partial f}{\partial v}\right)+\frac{1}{v^{2}}\frac{\partial }{\partial \mu }\left(D_{\mu \mu }\frac{\partial f}{\partial \mu }\right), \end{array}$$
where the right-hand side of the kinetic equation is expressed in a velocity-space spherical coordinate system, in which $v=\sqrt{v_{⊥}^{2}+v_{\|}^{2}}$ is the magnitude and $\mu =v_{\|}/v$ is the cosine of the pitch angle. Under the assumption of primarily electrostatic interaction, the velocity space friction and diffusion coefficients are given by
(11)AvAμ=e2km22πme∫dk∫dωIm1kϵ‖(k,ω)*μ1−μ2δ(ω−kvμ),DvvDvμDμμ=πe2me2∫dk∫dω〈δE‖2〉k,ωμ2μ(1−μ2)(1−μ2)2δ(ω−kvμ).
We assume steady state, ∂/∂t→0, and isotropy, ∂f/∂μ=0. Then, we average over μ. Then, we obtain the steady-state solution for the electron velocity distribution function,
(12)$$\begin{array}{ccc} f & = & \mathrm{const}\exp\left(-\int^{v}dv^{\prime }\;\frac{A(v^{\prime })}{D(v^{\prime })}\right),\\ A(v) & = & \int_{-1}^{1}d\mu A_{v},\;D\left(v\right)=\int_{-1}^{1}d\mu D_{vv}. \end{array}$$
We should note that this type of steady-state solution of the Fokker–Planck equation is found in the literature [66,67,68], so the basic concept is not new. Making use of the property Imϵ‖(k,ω)−1*=12πωpeδ(ω−ωL) and expressing ${\langle \delta E_{\|}^{2}\rangle }_{k,\omega }=I_{L}\left(k\right)\;\delta \left(\omega -\omega_{L}\right)$, where $I_{L}\left(k\right)$ is given by Equation (Equation 9), then we have
(13)$$A=\frac{e^{2}k_{m}^{2}}{2m_{e}}\int_{0}^{1}d\mu \mu ,\;D=\frac{T_{e}}{m_{e}v}\frac{e^{2}k_{m}^{2}}{2m_{e}}\left(\int_{0}^{1}d\mu \mu +\frac{k_{L}^{2}}{\omega_{pe}^{2}}\;v^{2}\int_{0}^{1}d\mu \mu^{3}\right).$$
From this, we obtain the desired electron kappa velocity distribution function,
(14)$$f=\mathrm{const}\;\exp\left(-\frac{m_{e}}{T_{e}}\int^{v}dv^{\prime }\;\frac{v^{\prime }}{1+k_{L}^{2}{v^{\prime }}^{2}/\left(2\omega_{pe}^{2}\right)}\right)=\mathrm{const}{\left(1+\frac{v^{2}}{\kappa \alpha^{2}}\right)}^{-\kappa },$$
if we identify
(15)$$\frac{k_{L}^{2}\alpha^{2}}{2\omega_{pe}^{2}}=\frac{1}{\kappa },\;\mathrm{and}\;\alpha^{2}=\frac{\kappa +1}{\kappa }\frac{2T_{e}}{m_{e}}.$$
In this version of the theory, the formation of a non-thermal (kappa) electron distribution is attributed to the Langmuir turbulence in the asymptotical steady state. According to this theory, no clear separation of the core and halo electrons is made, but, instead, both populations are treated as a single kappa distribution function with the low end of the velocity spectrum mimicking the Maxwellian thermal core, while the suprathermal high-velocity regime represents the inverse power-law tail population. The brief overview of this section is not new, and a full discourse can be found in Refs. [27,37,53,54,55,56,57,58,59,60,61,62,63,64,65]. In the remaining part of the present manuscript, we put forth a new model for which the role of whistler turbulence is emphasized.

## 4. Formation of Non-Thermal Electron Distribution by Combined Background Turbulence and Thermal Fluctuations

Section 2 discussed the thermal fluctuations spontaneously emitted by Maxwellian core electrons. We also discussed the effects of pre-existing solar wind turbulence and how the combined model may relate to the existing literature on low-frequency turbulence simulations. We also discussed how the effects of baseline quasi-thermal spontaneous emission fluctuations may impact the observations, although we noted that the unambiguous identification of the predicted spectral features associated with thermal fluctuations in the observation may depend on the level of turbulence. In this section, we proceed to discuss the combined impact of the quasi-thermal whistler-mode fluctuations and the background turbulence on the electron velocity distribution function.

In Section 3, the steady-state electron distribution function subject only to the Langmuir turbulence was discussed, and it was shown that the result is the kappa electron velocity distribution. For the kappa model, however, no distinction is made between the core and halo populations. Moreover, the spontaneously emitted transverse fluctuations in the whistler-mode frequency are ignored. Further, the presence of background solar wind turbulence is not taken into consideration either. As such, the kappa distribution and Langmuir turbulence problem may pertain to the outer heliosphere where the local ambient magnetic field strength is sufficiently low so that the whistler-mode frequency range effects can be ignored and the underlying plasma may be treated as essentially unmagnetized.

For the near-Earth space environment, however, the whistler-mode dynamics may be an integral part of wave–particle interaction with the electrons [32,33]. Thus, this section discusses the formation of the non-thermal electron velocity distribution function in the presence of spontaneous thermal fluctuations in both the longitudinal Langmuir and transverse whistler modes and also under the influence of background turbulence [52,69,70,71,72,73,74]. As will be shown, under such a physical environment, the self-consistent steady-state solution for the electron velocity distribution function will be characterized by a distinct core and halo populations, which is consistent with observations.

The notion of the pre-existing whistler-mode turbulence affecting the solar wind electron dynamics, resulting in a non-thermal velocity distribution function, has been discussed in the literature. For instance, Refs. [75,76,77,78] carried out extensive and detailed numerical simulation based on the quasilinear velocity diffusion theory where the diffusion coefficient is computed from the model whistler wave turbulence spectrum. It was shown in these references that the resonant wave–particle interaction between the solar wind electrons and the pre-existing turbulence in the whistler-mode frequency range leads to the gradual formation of a non-thermal energetic tail. The present paper is similar in conceptual background in that we are also seeking to find a non-thermal electron velocity distribution function that is a result of resonant wave–particle interaction with the background whistler wave turbulence. However, the main distinction between the present approach and those of previous works is twofold. Firstly, unlike the previous works, which relate to the dynamical evolution of the velocity distribution function, we are concerned with the steady-state solution. This aspect leads to the second distinction. That is, in order to obtain the steady-state solution, the effects of spontaneous thermal fluctuation are essential. The effects of spontaneous thermal fluctuations and the related velocity friction effects are not considered in the dynamical theories of solar wind electron distribution function in the above references. For dynamical problems, the velocity friction, which is intimately related to the spontaneous thermal fluctuations, is indeed relatively unimportant, but for the theory of an asymptotic steady state, the balance of velocity friction coefficient A and velocity diffusion coefficient D is crucially important—see Equation (Equation 12). Reference [67], however, considered a steady-state solution of a magnetospheric electron velocity distribution in resonant wave–particle interaction with the background whistler-mode waves. Their solution is very similar in conceptual background and mathematical methodology to the present work, except that, in their approach, the velocity friction coefficient is replaced by the collisional drag coefficient.

We again start from the Fokker–Planck electron kinetic equation with waves and fluctuations where a wave vector lying in the parallel direction is defined with the ambient magnetic field vector, that is, Equation (Equation 10) or the steady-state solution (Equation 12), except that now the electrons are immersed in the bath of thermal fluctuations of both Langmuir and whistler types and also the background turbulence. As a result, the velocity friction and the diffusion coefficients now contain contributions from both longitudinal and transverse modes,
(16)AvAμ=e2km24πme∫dk∫dωIm1ω2ϵ⊥(k,ω)−c2k2*×ωkv−ωμv(1−μ2)δ(ω−kvμ−Ωe)+e2km22πme∫dk∫dωIm1kϵ‖(k,ω)*μ1−μ2δ(ω−kvμ),DvvDvμDμμ=πe24me2∫dk∫dω〈δE⊥2〉k,ωω2(kv−ωμ)ω(kv−ωμ)2×1−μ2ω2δ(ω−kvμ−Ωe)+πe2me2∫dk∫dω〈δE‖〉k,ωμ2μ(1−μ2)(1−μ2)2δ(ω−kvμ).

For a steady state, the formal solution (Equation 12) is still applicable, with the coefficient $A=\int_{-1}^{1}d\mu A_{v}$ and $D=\int_{-1}^{1}d\mu D_{vv}$ now containing the influence of whistler-mode fluctuations as well as the background turbulence. Making use of Im[ω2ϵ⊥(k,ω)−c2k2]−1*=πδ(ω−ωW)(Ωe/ωpe2) and ${\langle \delta E_{⊥}^{2}\rangle }_{k,\omega }=I_{W}\left(k\right)\;\delta \left(\omega -\omega_{W}\right)$ and approximating the resonance delta function by $\delta \left(\omega_{W}-kv\mu -\Omega_{e}\right)\approx \delta \left(kv\mu +\Omega_{e}\right)$, we may proceed with the computation of generalized *A* and *D* coefficients. For the present purpose, we adopt the whistler-mode spectrum by considering the effects of thermal fluctuations and the background turbulence spectrum, that is, Equation (Equation 8), but in a simplified form. In particular, we are interested in the frequency range that is sufficiently higher than both the MHD scale and proton kinetic scale but is sufficiently below the electron cyclotron frequency. Thus, in such a low-frequency limit relative to the electron cyclotron frequency, $\omega^{2}\ll \Omega_{e}^{2}$, the whistler-mode fluctuations can be approximated by $I_{W}\left(\omega \right)\approx \left[k_{m}^{2}T_{e}/\left(4\pi \right)\right]\left(\omega \Omega_{e}/\omega_{pe}^{2}\right)$ and $M_{W}\left(k\omega \right)\approx \left[k_{m}^{2}T_{e}/\left(4\pi \right)\right]$, which, upon making use of the low-frequency version of the dispersion relation, $\omega =\Omega_{e}{\left(ck/\omega_{pe}\right)}^{2}$, can be written as
(17)$$I_{W}\left(k\right)∼\frac{k_{m}^{2}T_{e}}{4\pi }\frac{\Omega_{e}^{2}}{\omega_{pe}^{2}}\frac{c^{2}k^{2}}{\omega_{pe}^{2}},\;M_{W}\left(k\right)∼\frac{k_{m}^{2}T_{e}}{4\pi }.$$
This result, together with the Langmuir fluctuation spectrum, $I_{L}\left(k\right)∼k_{m}^{2}T_{e}/\left(4\pi \right)$, will be inserted into the expressions for *A* and *D*.

We may also simplify the model of the turbulence given by Equation (Equation 8) in that we only focus on the portion of the background turbulence spectrum corresponding to the whistler-mode range inverse power law, which we simplify by $\propto k^{-2\nu }$. If we thus superpose this simplified background turbulence spectrum to the approximate form of the spontaneous emission spectrum corresponding to the low-frequency whistler-mode thermal emission, then we may adopt a simplified form of the combined spectrum,
(18)$$I_{W}\left(k\right)=\frac{k_{m}^{2}T_{e}}{4\pi }\frac{\Omega_{e}^{2}}{\omega_{pe}^{2}}\left[\frac{c^{2}k^{2}}{\omega_{pe}^{2}}+{\left(\frac{k_{W}^{2}}{k^{2}}\right)}^{\nu }\right].$$
Here, $k_{W}^{2}$ is an appropriate parameter for correct dimensionality, which can be adjusted. This parameter effectively dictates the level of background turbulence as well. In applying the above model, we reiterate that the model spectrum (Equation 18) is meant for the whistler-mode frequency range satisfying $\Omega_{i}^{2}<\omega^{2}<\Omega_{e}^{2}$. As such, we confine the width of wave numbers, $c\Delta k/\omega_{pe}$, roughly corresponding to the above frequency limitation. In an earlier attempt to incorporate the solar wind turbulence effects into the model whistler-mode spectrum, Ref. [41] adopted a model where the thermal fluctuation spectrum was modified to reflect the inverse power-law feature, namely, $I_{W}\left(k\right)\to \left[k_{m}^{2}T_{e}/\left(4\pi \right)\right]\left(\Omega_{e}^{2}/\omega_{pe}^{2}\right){\left(c^{2}k^{2}/\omega_{pe}^{2}\right)}^{1-\beta }$, where β is a control parameter that can be chosen as 0 in the case of purely spontaneous emission and as β=1+ν if we wish to model the overall spectral profile to behave as an inverse power law, $I_{W}\left(k\right)\propto k^{-2\nu }$. However, we now realize that the more proper way to model the combined spontaneously emitted quasi-thermal whistler-mode spectrum and the background pre-existing turbulence is the linear superposition (Equation 8), which we simplify as shown in Equation (Equation 18). Thus, in the present section, we take the total whistler-mode spectral intensity to possess the proportionality dictated by the functional relationship, $I_{W}\left(k\right)\propto c^{2}k^{2}/\omega_{pe}^{2}+{\left(k_{W}^{2}/k^{2}\right)}^{\nu }$.

For the Langmuir mode spectrum, however, we only consider the thermal fluctuation, which is distinct from the previous section. Recall that in Section 3 we included the steady-state Langmuir turbulence factor, ${\left(k_{L}/k\right)}^{2}$, in the Langmuir turbulence spectrum (Equation 9), which led to the electron kappa distribution. In the present section, we are concerned with an alternative theory of a non-thermal, generalized kappa distribution, which is based upon the notion of background whistler-mode turbulence. In short, the transverse and longitudinal electric field spectral intensities adopted in the present discussion are defined by
(19)$$\begin{array}{ccc} \frac{{\langle \delta E_{⊥}^{2}\rangle }_{k,\omega }}{4} & = & \frac{k_{m}^{2}T_{e}}{4\pi }\frac{\Omega_{e}^{2}}{\omega_{pe}^{2}}\left[\frac{c^{2}k^{2}}{\omega_{pe}^{2}}+{\left(\frac{k_{W}^{2}}{k^{2}}\right)}^{\nu }\right]\delta \left(\omega -\omega_{W}\right),\\ {\langle \delta E_{\|}^{2}\rangle }_{k,\omega } & = & \frac{k_{m}^{2}T_{e}}{4\pi }\;\delta \left(\omega -\omega_{L}\right). \end{array}$$
Inserting this into the generalized coefficients (Equation 16), we obtain the desired coefficients *A* and *D*, which are now given by
(20)$$\begin{array}{ccc} A & = & \frac{e^{2}k_{m}^{2}}{2m_{e}}\left(\frac{\Omega_{e}^{4}}{\omega_{pe}^{4}}\frac{c^{2}}{v^{2}}\int_{0}^{1}d\mu \;\frac{1-\mu^{2}}{\mu^{3}}+\int_{0}^{1}d\mu \mu \right),\\ D & = & \frac{T_{e}}{m_{e}v}\frac{e^{2}k_{m}^{2}}{2m_{e}}\frac{\Omega_{e}^{2}}{\omega_{pe}^{2}}\left[\frac{c^{2}}{v^{2}}\frac{\Omega_{e}^{2}}{\omega_{pe}^{2}}\int_{0}^{1}d\mu \;\frac{1-\mu^{2}}{\mu^{3}}+{\left(\frac{k_{W}^{2}}{\Omega_{e}^{2}}\right)}^{\nu }v^{2\nu }\int_{0}^{1}d\mu \left(1-\mu^{2}\right)\mu^{2\nu -1}\right]\\ & & +\frac{T_{e}}{m_{e}v}\frac{e^{2}k_{m}^{2}}{2m_{e}}\int_{0}^{1}d\mu \mu . \end{array}$$
The integral $\int_{0}^{1}d\mu \;\left(1-\mu^{2}\right)\mu^{-3}$ is formally divergent. To regularize the divergence, we introduce the lower limit, $\int_{0}^{1}d\mu \;\left(1-\mu^{2}\right)\mu^{-3}\to \int_{\mu_{\min}}^{1}d\mu \;\left(1-\mu^{2}\right)\mu^{-3}=\left(1-\mu_{\min}^{2}\right)/\left(2\mu_{\min}^{2}\right)+\ln\mu_{\min}$. The other μ integral is evaluated in a straightforward manner: $\int_{0}^{1}d\mu \left(1-\mu^{2}\right)\mu^{2\nu -1}=1/\left[2\nu \left(\nu +1\right)\right]$. Making use of all this, we have
(21)$$\begin{array}{ccc} f & = & C\exp\left(-\int^{x}dx^{\prime }\;\frac{2x^{\prime }\left(a+{x^{\prime }}^{2}\right)}{a\left[1+{\left({x^{\prime }}^{2}/\kappa_{W}\right)}^{\nu +1}\right]+{x^{\prime }}^{2}}\right),\;x=\frac{v}{\alpha_{e}},\\ a & = & \frac{2\Lambda }{{\left(\omega_{pe}/\Omega_{e}\right)}^{2}\beta_{e}},\;\kappa_{W}=\frac{1}{\beta_{e}}{\left(\frac{2\nu (\nu +1)\Lambda }{{\left(ck_{W}/\Omega_{e}\right)}^{2\nu }}\right)}^{\frac{1}{\nu +1}},\\ \Lambda & = & \frac{1-\mu_{\min}^{2}}{2\mu_{\min}^{2}}+\ln\mu_{\min}. \end{array}$$
Here, we have made use of ${\left(c/\alpha_{e}\right)}^{2}{\left(\Omega_{e}/\omega_{pe}\right)}^{2}=B_{0}^{2}/\left(8\pi n_{0}T_{e}\right)=1/\beta_{e}$. This is a three-parameter model distribution, with ν, *a*, and $\kappa_{W}$ being the adjustable parameters. If we consider the limit of a→∞, then we have
(22)$$f_{W\mathrm{only}}=C\exp\left(-2\int^{x}dx^{\prime }\;\frac{x^{\prime }}{1+{\left({x^{\prime }}^{2}/\kappa_{W}\right)}^{\nu +1}}\right).$$
In this limit, the contribution from electrostatic Langmuir-mode fluctuation, that is, the term ${x^{\prime }}^{2}$ in both the numerator and denominator within the integrand, is ignored. This limiting form can be termed the *W*-only distribution. However, if we take the limit of a→0, then we simply have
(23)$$f_{L\mathrm{only}}=C\exp\left(-2\int^{x}x^{\prime }\;dx^{\prime }\right)=C\exp\left(-x^{2}\right),$$
the Maxwell–Boltzmann (MB) distribution. In this limit, the contribution from the whistler-mode related terms are ignored, and, thus, this limit can be termed the *L*-only distribution. Another interesting limit is when $\kappa_{W}\to \infty$. In this limit, the contribution from the background turbulence disappears, and the resulting distribution is that of the MD distribution again.

The parameters *a* and $\kappa_{W}$, in turn, are determined by $\mu_{\min}$, $\omega_{pe}/\Omega_{e}$, $\beta_{e}$, ν, and $ck_{W}/\Omega_{e}$. The parameter $\omega_{pe}^{2}/\Omega_{e}^{2}$ can be determined from the solar wind data. Also, $\beta_{e}$ is known from the data. The fitting parameters $ck_{W}/\Omega_{e}$ and ν relate to the spectral profile of the solar wind turbulence in the whistler-mode frequency range. Thus, these parameters can also be determined from observational properties. The truly free parameter is Λ, which is determined from the choice of $\mu_{\min}$. Let us consider the resonance condition, $kv\mu +\Omega_{e}=0$ or $\mu =-\Omega_{e}/\left(kv\right)$. We are interested in the minimum value for μ. In the formal μ integral, this is taken to be $\mu_{\min}=0$, but this means either k→∞ or v→∞, neither of which are physical. For whistler turbulence and fluctuations in the low-frequency limit, we choose the maximum *k* by $ck_{\max}/\omega_{pe}∼1$ or so. For the velocity *v*, we generally determine the maximum value to be sufficiently higher than the thermal speed, $v_{\max}\gg \alpha_{e}$. From this, we may see that $\mu_{\min}∼-\Omega_{e}/\left(k_{\max}v_{\max}\right)$, which, while small, can have a substantial range of freedom. If, for instance, we choose $\mu_{\min}∼10^{-6}$ or so, then we obtain $\Lambda ∼10^{10}$. However, if we choose $\mu_{\max}$ to be approximately $10^{-4}$ or so, then we have $\Lambda ∼10^{6}$ and so on and so forth. With this information, let us consider the ratio $a/\kappa_{W}$. If we choose ν=1, which implies $k^{-2}$ spectral behavior associated with the whistler frequency range turbulence, then we have
(24)$$\frac{a}{\kappa_{W}}=\frac{\left(ck_{W}/\Omega_{e}\right)}{{\left(\omega_{pe}/\Omega_{e}\right)}^{2}\beta_{e}}\;\Lambda^{1/2}.$$
Suppose we take ${\left(\omega_{pe}/\Omega_{e}\right)}^{2}\beta_{e}=10^{2}$ and $ck_{W}/\Omega_{e}=0.2$. Then, by choosing $\Lambda =2.25\times 10^{6}$ or so, we arrive at $a/\kappa_{W}∼3$. If, however, ${\left(\omega_{pe}/\Omega_{e}\right)}^{2}\beta_{e}=10^{4}$, then the choice of $\Lambda =2.25\times 10^{10}$ leads to the similar value of $a/\kappa_{W}∼3$. In the solar wind, the ratio $\omega_{pe}/\Omega_{e}$ can be quite high, ranging from O(10) to $O(10^{2})$. The electron beta value in the solar wind can range from $\beta ∼O(10^{-2})$ to O(1) or so, hence the above two choices of parameters, ${\left(\omega_{pe}/\Omega_{e}\right)}^{2}\beta_{e}=10^{2}$ and ${\left(\omega_{pe}/\Omega_{e}\right)}^{2}\beta_{e}=10^{4}$. The choice of $ck_{W}/\Omega_{e}=0.2$ relates to the turbulence property in that this number represents the maximum effective range of whistler-mode turbulence in the wave number space. Since the low-frequency whistler mode is characterized by $ck_{W}/\Omega_{e}<1$, such a choice is eminently reasonable. The above estimation of the crucial dimensionless parameter $a/\kappa_{W}$, of course, is a rough exercise, and more precise attempts should be made by surveying the 1 au data. However, in view of the uncertainty associated with the lower limit of the cosine of the pitch angle, $\mu_{\min}$, we defer the more accurate attempts for future.

With these considerations, we construct the asymptotic electron velocity distribution function (Equation 21) by performing a numerical indefinite integration over the dimensionless velocity $x=v/\alpha_{e}$. The result is displayed in Figure 6, where we display on the left-hand panel the case for $a/\kappa_{W}=3$ with a=30 and $\kappa_{W}=10$. For all the examples, we restrict ourselves to ν=1. As visual guides, we plot the so-called *L*-only and *W*-only limiting case distributions. We also plot the inverse power-law velocity slop $v^{-6.5}$. In the solar wind, such an asymptotic high-velocity tail distribution is often observed [29]. On the right-hand panel, we show the velocity distribution by varying the parameter *a*, which ranges from a=1 to 10 to 20 to 30. Other parameters are fixed: $\kappa_{W}=10$ and ν=1. Figure 6 thus demonstrates that the combined effects of background turbulence and finite spontaneously emitted fluctuations are capable of producing the electron velocity distribution function that remarkably resembles the observed distribution in the solar wind. We should note, however, that there exists a certain degree of freedom in our choice of parameters *a* and $\kappa_{W}$, in particular, their ratio, $a/\kappa_{W}$, which turns out to be important for determining the shape of the velocity distribution function, as the right-hand panel of Figure 6 indicates.

## 5. Summary

The main purpose of the present paper was to put forth a first principle theory of the steady-state electron velocity distribution function with non-thermal characteristics, which resembles the quiet-time solar wind electron distribution detected in the near-Earth space environment. Unlike the previous model [27], which invoked the steady-state Langmuir turbulence and the accompanying kappa distribution, the present paper employed the combined influence of the background solar wind turbulence in the whistler frequency range, as well as the quasi-thermal electromagnetic and electrostatic fluctuations. The resulting model electron distribution function was given in terms of an exponential function of an indefinite velocity integral, which does not in general lend itself to further closed-form analytical manipulations—Equation (Equation 21)—but must, in general, be computed by numerical means. Under a reasonable set of assumptions and input conditions, we have found that the resulting numerical calculation leads to a velocity distribution function whose profile is reminiscent of the measured distribution in space.

The formal electron velocity distribution function in the steady-state was given by Equation (Equation 21), and this mathematical expression contains the effects of background fluctuations as well as the impact of the pre-existing whistler-mode turbulence. This contrasts to the formal solution (Equation 14), which reflects the influence of electrostatic thermal fluctuations and the enhanced Langmuir wave turbulence. It is possible to construct a more general formal distribution by combining the effects of both the electrostatic and electromagnetic thermal fluctuations on the electrons, as well as the influence of enhanced electrostatic (Langmuir) and electromagnetic (whistler) turbulence intensities. The result is
(25)$$f\left(x\right)=C\exp\left(-\int^{x}dx^{\prime }\;\frac{2x^{\prime }\left(a+{x^{\prime }}^{2}\right)}{a\left[1+{\left({x^{\prime }}^{2}/\kappa_{W}\right)}^{\nu +1}\right]+\left(1+{x^{\prime }}^{2}/\kappa_{L}\right){x^{\prime }}^{2}}\right),\;x=\frac{v}{\alpha_{e}}.$$
Here, $\kappa_{L}=2\omega_{pe}^{2}/{\left(k_{L}\alpha \right)}^{2}$, as defined in Equation (Equation 15). With this form of the electron distribution function, we now summarily discuss the various limits. Suppose that we ignore the influence of background turbulence altogether. This amounts to taking $\kappa_{W}\to \infty$ and $\kappa_{L}\to \infty$, which leads to
(26)$$f_{\mathrm{MB}}\left(x\right)=C\exp\left(-x^{2}\right)\;\left(\kappa_{W}\to \infty ,\kappa_{L}\to \infty \right).$$
Thus, in the absence of turbulence, we obtain the MB distribution, which is as expected. Ignoring the influence of electromagnetic whistler modes, both the thermal fluctuations and the enhanced pre-existing turbulence, is equivalent to taking the limit of a→0, which leads to the generalized form of the *L*-only distribution—see Equation (Equation 23)—which also happens to be the same kappa distribution discussed in Equation (Equation 14)
(27)$$f_{L\mathrm{only}}\left(x\right)=f_{\kappa }\left(x\right)=C{\left(1+\frac{x^{2}}{\kappa_{L}}\right)}^{-\kappa_{L}}\;\left(a\to 0\right).$$
If, however, we are to ignore the influence of electrostatic modes, both the thermal fluctuations and enhanced Langmuir turbulence, then all we need to do is consider the limit of a≫1,
(28)$$f_{W\mathrm{only}}\left(x\right)=C\exp\left(-\int^{x}dx^{\prime }\;\frac{2x^{\prime }}{1+{\left({x^{\prime }}^{2}/\kappa_{W}\right)}^{\nu +1}}\right)\;\left(a\gg 1\right).$$
This form of limiting solution was referred to as the *W*-only distribution in Equation (Equation 22), but this stand-alone solution is not a realistic model since the electrostatic fluctuations cannot simply be ignored. Nevertheless, at least as a mathematical exercise, one can certainly entertain such a limit. In our final solution (Equation 21), we have considered the limit of $\kappa_{L}\to \infty$, while other parameters, *a* and $\kappa_{W}$, are considered finite.

The overall concept of charged particles maintaining a steady-state wave–particle interaction with steady-state turbulence and fluctuations is an example of a stationary state far from equilibrium [79]. Such a state, in turn, may be considered as an example of the non-extensive statistical state [18,19]. It is in this regard that the present paper is relevant to the Special Issue “Nonadditive Entropies and Nonextensive Statistical Mechanics”. The fact that the space plasma, which is governed by a long-range electromagnetic force, frequently exhibits a kappa-like non-thermal distribution function is in a way not too surprising in that, thanks to Tsallis’s pioneering work, we now have a rather insightful understanding that any system with a long-range interaction is likely to be governed by a non-extensive, non-additive statistical principle. The present paper, as with the related earlier work [27], provides the physical “mechanism” that leads to a concrete example of a kappa-like non-thermal phase space distribution function.

## Figures and Tables

**Figure 1 entropy-26-00310-f001:**
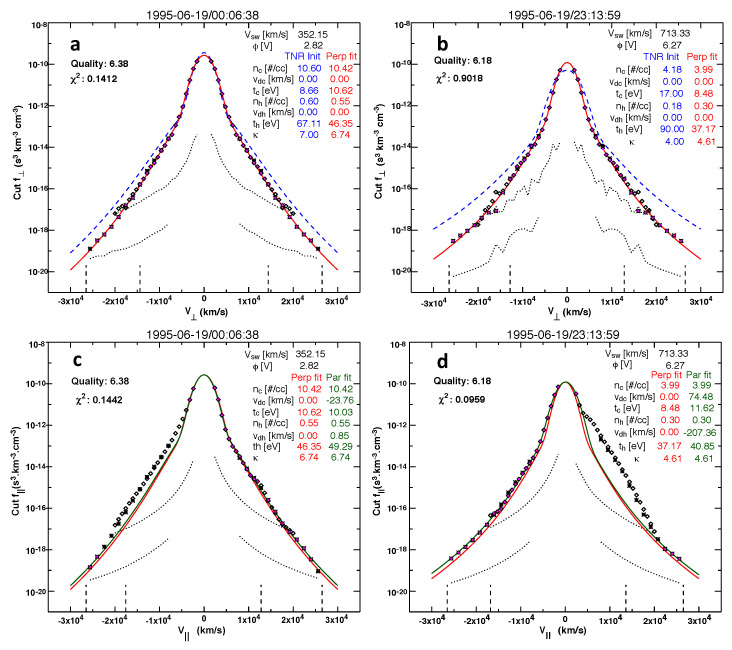
Reproduced from Figure 4 of Ref. [11]: Two typical electron velocity distribution functions (eVDFs) measured by EESA-L and EESA-H onboard Wind spacecraft at 1 au in the slow solar wind—panels (**a**,**c**)—and in the fast solar wind—panels (**b**,**d**). The top panels (**a**,**b**) show cuts through the eVDF in one of the two directions perpendicular to the local magnetic field *B*. The bottom panels (**c**,**d**) show cuts through the eVDF in the direction parallel to *B*. Explanations for the different lines in the figure is given in the main text. For more details, see Ref. [11].

**Figure 2 entropy-26-00310-f002:**
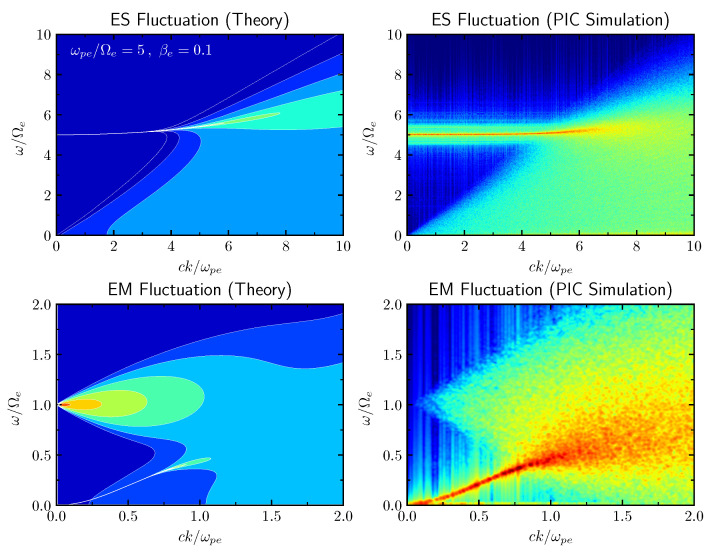
[**Upper-left**] Electrostatic fluctuation spectrum ${\langle \delta E_{\|}^{2}\rangle }_{k,\omega }$ computed from theory; [**upper-right**] simulated electrostatic fluctuation spectrum; [**lower-left**] electromagnetic fluctuation spectrum ${\langle \delta E_{⊥}^{2}\rangle }_{k,\omega }$ computed from theory; [**lower-right**] simulated electromagnetic fluctuation spectrum. These spectra are plotted as a function of $ck/\omega_{pe}$ (horizontal axis) and $\omega /\Omega_{e}$ (vertical axis), with their relative intensities indicated by color maps in arbitrary scales.

**Figure 3 entropy-26-00310-f003:**
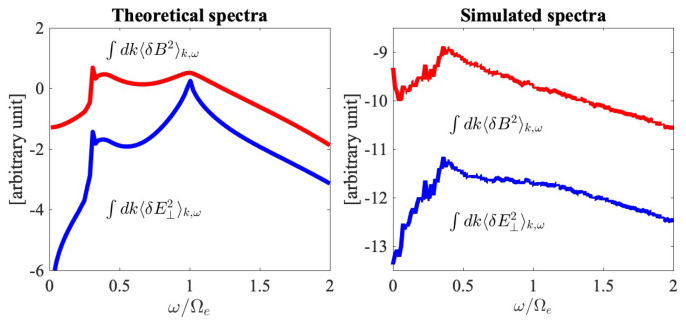
[**Left**] *k*-integrated magnetic and transverse electric field fluctuation spectra $\int dk{\langle \delta B^{2}\rangle }_{k,\omega }$ (red) and $\int dk{\langle \delta E_{⊥}^{2}\rangle }_{k,\omega }$ (blue) computed from theory. [**Right**] Simulated fluctuation spectra integrated over *k*. The integrated spectra are plotted against $\omega /\Omega_{e}$.

**Figure 4 entropy-26-00310-f004:**
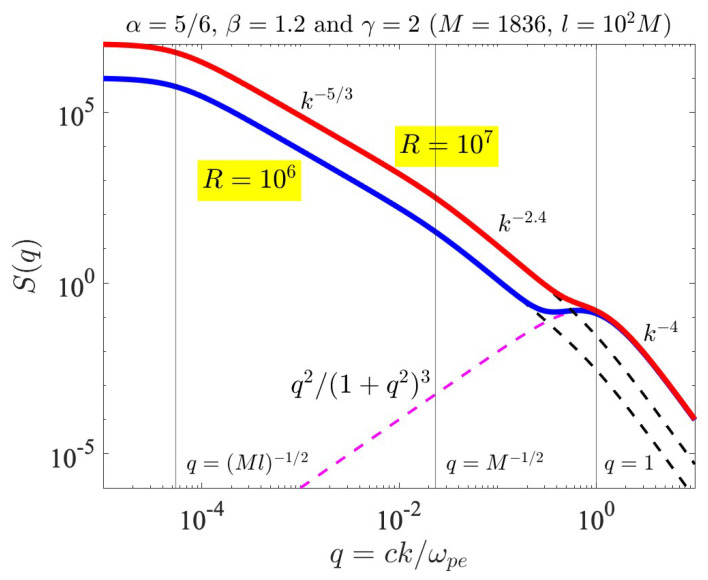
Model spectral factor, S(q), versus *q* in logarithmic horizontal and vertical scales. The magenta dashes represent the spontaneous whistler-mode fluctuation, while the black dashes denote the background turbulence for two cases of $R=10^{6}$ and $R=10^{7}$. The combined spectra are plotted with thick blue ($R=10^{6}$) and red ($R=10^{7}$) curves. For a low turbulence level ($R=10^{6}$), the presence of fluctuation should be more evident, but if the turbulence level is high ($R=10^{7}$), then the fluctuation will be partially hidden.

**Figure 5 entropy-26-00310-f005:**
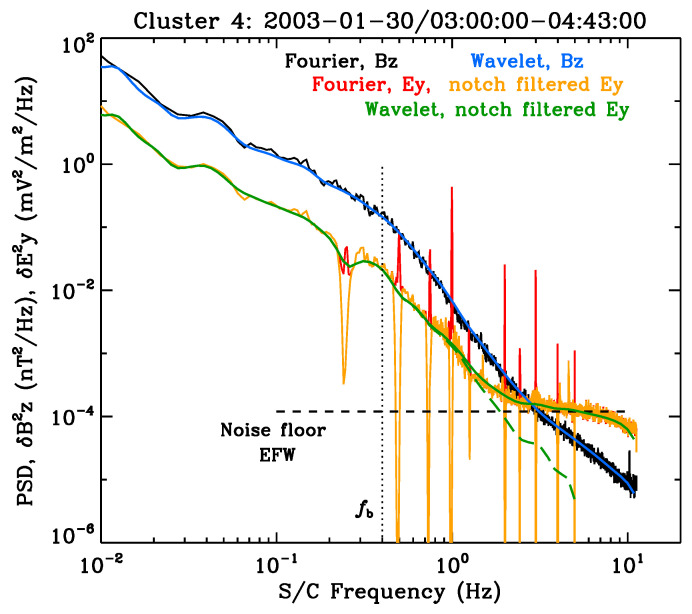
Magnetic (black) and electric (red) field spectra taken by *Cluster* spacecraft [reproduced from Ref. [52]].

**Figure 6 entropy-26-00310-f006:**
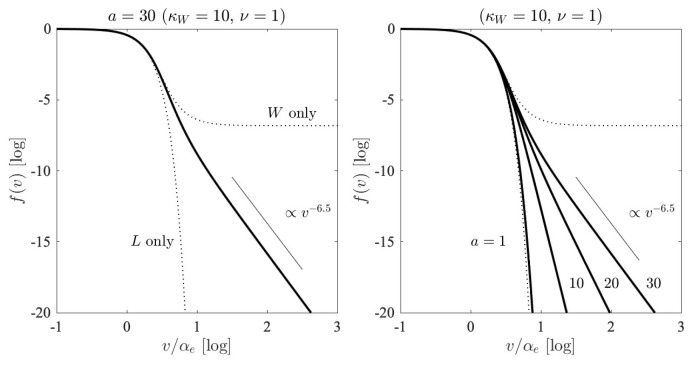
[**Left**] The three-parameter electron velocity distribution function *f* versus $x=v/\alpha_{e}$, for a=30, $\kappa_{W}=10$, and ν=1. [**Right**] The variation in the input parameter *a*, which ranges from a=1 to 10 to 20 to 30.

## Data Availability

Data are contained within the article.

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
