# Peer review of "Non-Thermal Solar Wind Electron Velocity Distribution Function"

_entropy, 2024, doi:10.3390/e26040310_

Round 1

Reviewer 1 Report

Comments and Suggestions for Authors

Review report Yoon et al. 2024 Non-thermal solar-wind electron velocity distribution function

This paper mainly reminds the principles of a previous model developed by the main author, which involves the electrostatic turbulence near the plasma oscillation frequency (i.e., Langmuir turbulence), and in addition considers the impact of transverse electromagnetic turbulence in the whistler-mode frequency range. The part concerning whistler turbulence constitutes thus the main innovative part, but it is incomplete for the context and bibliography. It should better present previous works and simulations made on this topic. Indeed, the first suggestions and simulations showing that whistler turbulence can be at the origin of Kappa-like distributions are much older than the present paper and the references that are provided. Let us cite for instance (and references inside):

Vocks and Mann, Generation of Suprathermal Electrons by Resonant Wave-Particle Interaction in the Solar Corona and Wind, ApJ, 593, 1134, DOI 10.1086/376682, 2003

Pierrard V., M. Lazar and R. Schlickeiser. Evolution of the electron distribution function in the whistler wave turbulence of the solar wind, Solar Phys. 269, 2, 421-438, DOI 10.1007/s11207-010-9700-7, 2011

The principles and the equations used to take into account the different turbulences are provided in detail, but the simulations themselves are less clearly explained. The authors admit that they take arbitrary values for the parameters to illustrate the effects. I think that the paper could be improved by a more detailed justification of the choices, especially considering the new observations of Parker Solar Probes concerning whistler waves.

Also, since the thermal fluctuations of both Langmuir and whistler types, and also the background turbulence are considered, would it be possible to explain in more detail in the conclusions what are the effects of each component independently and what is their relative importance as a function of the distance?

 Other corrections:

L54-56 Please explain in more detail what is the blue line in Figure 1 (sum of Maxwellian and Kappa obtained from a fit?)  and what is the difference with the red line.

L72 “ apparently were independently  discovered several times over as entry 107 in Ref. [18].” The reference 18 is from Tsallis while it should mention other author(s) if the discovery was already made by other scientist(s).

L99 Please specify the energy of superhalo in comparison to halo and strahl to well explain the differences.

L106 Only very recent references are cited concerning the generation of suprathermal tails by whistler turbulence, while the first works on this are much more ancient. Several references that showed this previously should be mentioned.

L110 Note that the fact that suprathermal tails are created, even with a power law, does not mean that it is really a kappa or a Tsallis distribution. A sum of Maxwellian + Kappa seems to give better fits at low distances since they can be more precise, depending on the position of the break point between thermal velocities and the formation of the power law tail. Even in the observations (as shown by WIND in Fig. 1), it is not fully a Kappa function, due to the anisotropy and the presence of the strahl for instance. It would be also interesting to determine what generates the tails for the strahl, and what is more due to the spread of strahl to halo.

L151 Same remarks as above. A kappa-like distribution or a power law tail? What about the anisotropy? What is new in comparison to the previous simulations from other authors that showed that whistlers generate Kappa tails?

Fig 3 Please comment more clearly why the theoretical peak at w/Omega_e=1 is not reproduced in the simulations. 

Fig. 5 L266 The spectral flattening behavior for high-frequency end of the turbulence spectra for the electric field E is not clearly significant due to the noise floor. This is mentioned, but only later in the text. Are there other measurements or arguments that also indicate this behavior?

Fig. 6: Would it be possible to better determine or justify the values of the parameters, especially with the new observations of Parker Solar Probes?

Typos:

L219 the Sun (upper case like elsewhere in the text)

L 230 to behave

L256 k^2.4 (not 2/4) to be coherent with the figure

L408 “also” 2 times in the same sentence

L460 may be

L464 Tsallis (add i)

Ref 28 steady state (not strady)

Author Response

Comments: This paper mainly reminds the principles of a previous model developed by the main author, which involves the electrostatic turbulence near the plasma oscillation frequency (i.e., Langmuir turbulence), and in addition considers the impact of transverse electromagnetic turbulence in the whistler-mode frequency range. The part concerning whistler turbulence constitutes thus the main innovative part, but it is incomplete for the context and bibliography. It should better present previous works and simulations made on this topic. Indeed, the first suggestions and simulations showing that whistler turbulence can be at the origin of Kappa-like distributions are much older than the present paper and the references that are provided. Let us cite for instance (and references inside):

Vocks and Mann, Generation of Suprathermal Electrons by Resonant Wave-Particle Interaction in the Solar Corona and Wind, ApJ, 593, 1134, DOI 10.1086/376682, 2003

Pierrard V., M. Lazar and R. Schlickeiser. Evolution of the electron distribution function in the whistler wave turbulence of the solar wind, Solar Phys. 269, 2, 421-438, DOI 10.1007/s11207-010-9700-7, 2011

Response: Yes, the reviewer is absolutely correct, so we now add these other references and also add an entirely new paragraph addressing the previous works, as well as to explain the distinctions and similarities between the previous works and our approach. We thank the reviewer for pointing this aspect out to us.

Comments (cont'd): The principles and the equations used to take into account the different turbulences are provided in detail, but the simulations themselves are less clearly explained. The authors admit that they take arbitrary values for the parameters to illustrate the effects. I think that the paper could be improved by a more detailed justification of the choices, especially considering the new observations of Parker Solar Probes concerning whistler waves.

Response: We now provide more details on the simulation setup including the ratio of plasma-to-gyro frequencies. We also note that our choice is somewhat arbitrary but also provide justification.

Comments (cont'd): Also, since the thermal fluctuations of both Langmuir and whistler types, and also the background turbulence are considered, would it be possible to explain in more detail in the conclusions what are the effects of each component independently and what is their relative importance as a function of the distance?

Response: This is an excellent suggestion, and we now do exactly that in the penultimate paragraph of the Summary section.

Other corrections:

L54-56 Please explain in more detail what is the blue line in Figure 1 (sum of Maxwellian and Kappa obtained from a fit?)  and what is the difference with the red line.

Response: We now add further explanations regarding Figure 1. See the highlighted portion in the text.

L72 ``apparently were independently discovered several times over as entry 107 in Ref. [18].'' The reference 18 is from Tsallis while it should mention other author(s) if the discovery was already made by other scientist(s).

Response: We agree with the reviewer. It would be more self-contained to specifically mention the early works for the benefit of the readers. Thus, we now recapitulate the content from entry 107 of Ref. [18] in the footnote, for the sake of completeness.

L99 Please specify the energy of superhalo in comparison to halo and strahl to well explain the differences.

Response: We now added more comments on the properties of these various distinct electron components.

L106 Only very recent references are cited concerning the generation of suprathermal tails by whistler turbulence, while the first works on this are much more ancient. Several references that showed this previously should be mentioned.

Response: The references cited in L106 only relate to the whistler waves. Other references that deal with the suprathermal tail generation, as suggested by the reviewer plus others are cited later - Refs. 75--78.

L110 Note that the fact that suprathermal tails are created, even with a power law, does not mean that it is really a kappa or a Tsallis distribution. A sum of Maxwellian + Kappa seems to give better fits at low distances since they can be more precise, depending on the position of the break point between thermal velocities and the formation of the power law tail. Even in the observations (as shown by WIND in Fig. 1), it is not fully a Kappa function, due to the anisotropy and the presence of the strahl for instance. It would be also interesting to determine what generates the tails for the strahl, and what is more due to the spread of strahl to halo.

Response: The reviewer is correct and we now clarify that the nonthermal VDF is not necessarily the Kappa model. In fact, our model VDF must be generated by numerically carrying out the indefinite velocity integral.

L151 Same remarks as above. A kappa-like distribution or a power law tail? What about the anisotropy? What is new in comparison to the previous simulations from other authors that showed that whistlers generate Kappa tails?

Response: We also made a similar modification.

Fig 3 Please comment more clearly why the theoretical peak at \omega/\Omega_e=1 is not reproduced in the simulations.

Response: We now add some comments on the discrepancy. It is owing to the issue related to the resolution of the higher-order mode in the simulation.

Fig. 5 L266 The spectral flattening behavior for high-frequency end of the turbulence spectra for the electric field E is not clearly significant due to the noise floor. This is mentioned, but only later in the text. Are there other measurements or arguments that also indicate this behavior?

Response: The reviewer is correct that the observed flattening behavior could be entirely due to instrument artifact, which we now mention.

Fig. 6: Would it be possible to better determine or justify the values of the parameters, especially with the new observations of Parker Solar Probes?

Response: At the present level of our theoretical effort, we believe that more accurate attempts to model the parameters are not terribly meaningful. This is because our theory depends on the arbitrary low-$\mu$ cutoff value in the divergent integral, $\int_0^1d\mu(1-\mu^2)\mu^{-3}\to\int_{\mu_{\rm min}}^1d\mu(1-\mu^2)\mu^{-3}$. We now mention this.

Typos:

L219 the Sun (upper case like elsewhere in the text)

L 230 to behave

L256 k^{2.4} (not 2/4) to be coherent with the figure

L408 ``also'' 2 times in the same sentence

L460 may be

L464 Tsallis (add i)

Ref 28 steady state (not strady) \\

Response: We thank the reviewer for pointing these typos out. They are corrected.

Reviewer 2 Report

Comments and Suggestions for Authors

This paper investigates the formation of suprathermal electron distributions, i.e., kappa distribution with power-law like tails, in the solar wind under the influence of thermal fluctuations and background turbulence originating from the Sun. In contrast to previous studies, it not only considers electrostatic Langmuir oscillations but also electromagnetic waves in the Whistler wave range.

Thermal fluctuation spectra are presented both from theoretical predictions and from numerical studies. Model spectra of solar wind turbulence are added and the resulting total electric and magnetic field spectra compared to observations.

In the subsequent chapter, velocity distribution functions calculated under the influence of this combined fluctuation spectrum. They reproduce typical solar wind features, like a thermal core and an extended, power-law halo.

The paper is generally well written. The theory and the relation of this manuscript to previous work is thoroughly explained. I just have a few minor comments:

- line 40 - 43:
It is stated that the kappa distribution approximates the Maxwellian for small v. Actually, from calculating the derivative of eq. (1) the kappa distribution then approximates a Maxwellian with a smaller thermal speed, by a factor sqrt(kappa / (kappa + 1))

- Discussion of Fig. 2, lines 182 - 191:
There are some distinct differences between the theoretical and numerical results. The theory predicts a maximum at omega = Omega_e and c  /omega_pe = 1, that is not present in simulations. Is this difference due to numerical issues, e.g. finite size of the simulation box?

- Discussion of Fig. 3, lines 200 - 207:
The theoretical E-field spectrum in Fig. 3 shows a local maximum at omega = Omega_e, that is not found in the simulation. This could be commented.

- Figure 2:
What is the color scale, and is it the same for all plots?

- lines 214 - 216:
This calculation is very concise, although well-known plasma theory. A reference would be helpful.

- Figure 4:
A text block in the figure caption is just a copy of the paragraph just above the figure.

- Formula for electron distribution, eqs. (12), (14):
How can the distribution function f, which should be a function of velocity, v, be calculated as in integral over v?

Author Response

Comments: This paper investigates the formation of suprathermal electron distributions, i.e., kappa distribution with power-law like tails, in the solar wind under the influence of thermal fluctuations and background turbulence originating from the Sun. In contrast to previous studies, it not only considers electrostatic Langmuir oscillations but also electromagnetic waves in the Whistler wave range.

Thermal fluctuation spectra are presented both from theoretical predictions and from numerical studies. Model spectra of solar wind turbulence are added and the resulting total electric and magnetic field spectra compared to observations.

In the subsequent chapter, velocity distribution functions calculated under the influence of this combined fluctuation spectrum. They reproduce typical solar wind features, like a thermal core and an extended, power-law halo.

The paper is generally well written. The theory and the relation of this manuscript to previous work is thoroughly explained. I just have a few minor comments:

- line 40 - 43: It is stated that the kappa distribution approximates the Maxwellian for small v. Actually, from calculating the derivative of eq. (1) the kappa distribution then approximates a Maxwellian with a smaller thermal speed, by a factor \sqrt(\kappa / (\kappa + 1))

Response: The reviewer is correct. We now make a note of this more accurate property.

- Discussion of Fig. 2, lines 182 - 191: There are some distinct differences between the theoretical and numerical results. The theory predicts a maximum at \omega = \Omega_e and ck/\omega_{pe} = 1, that is not present in simulations. Is this difference due to numerical issues, e.g. finite size of the simulation box?

Response: Yes, it is indeed related to the simulation not fully resolving the higher-order mode near \omega=\Omega_e, especially as k approaches 0. We now mention this.

- Discussion of Fig. 3, lines 200 - 207: The theoretical E-field spectrum in Fig. 3 shows a local maximum at \omega = \Omega_e, that is not found in the simulation. This could be commented.

Response: The other reviewer raised the same question, we addressed this issue, but it is indeed related to the simulation not fully resolving the higher-order mode near \omega=\Omega_e, especially as k approaches 0.

- Figure 2: What is the color scale, and is it the same for all plots?

Response: It is relative for each panel, as we now explain.

- lines 214 - 216: This calculation is very concise, although well-known plasma theory. A reference would be helpful.

Response: We now cite the relevant literature -- Refs. [39, 40, 79].

- Figure 4: A text block in the figure caption is just a copy of the paragraph just above the figure.

Response: We are not entirely certain what the reviewer is saying, but in case he/she is pointing out the repetition of the text and the caption, we have slightly changed the figure caption to avoid the repetition.

- Formula for electron distribution, eqs. (12), (14): How can the distribution function f, which should be a function of velocity, v, be calculated as in integral over v?

Response: We now reformat the indefinite velocity integral to address the reviewer's comment. We thank the reviewer for pointing this potentially confusing issue out.

Round 2

Reviewer 1 Report

Comments and Suggestions for Authors

I thank the authors for their clear answers. I think that their additions in the article highly improve the manuscript. I strongly recommend it for publication. I have just some very minor edition suggestions that the authors can consider:

L30: Replace – by a final point, it will be clearer.

L53 offers

L59 please put (see equation (1)) in parentheses instead of – (again to avoid to mix with minus sign)

L175 is too high (add “is”)